# Implications of reducing antibiotic treatment duration for antimicrobial resistance in hospital settings: A modelling study and meta-analysis

Yin Mo[1,2,3,4]* , Mathupanee Oonsivilai[1,2], Cherry Lim[1,2], Rene Niehus[5], Ben S. Cooper[1,2]

1 Centre for Tropical Medicine and Global Health, Nuffield Department of Medicine, University of Oxford, Oxford, United Kingdom, 2 Mahidol-Oxford Tropical Medicine Research Unit, Faculty of Tropical Medicine, Mahidol University, Bangkok, Thailand, 3 Division of Infectious Diseases, University Medicine Cluster, National University Hospital, Singapore, Singapore, 4 Department of Medicine, National University of Singapore, Singapore, Singapore, 5 Harvard T.H. Chan School of Public Health, Harvard University, Boston, Massachusetts, United States of America

☯ These authors contributed equally to this work.
* moyin@tropmedres.ac

⊇ OPEN ACCESS

**Data Availability Statement:** The computer code for all simulations and data for the meta-analysis are publicly available (https://github.com/moyinNUHS/abxduration_abm).

**Funding:** This work was supported by the Singapore National Medical Research Council

## Abstract

### Background

Reducing antibiotic treatment duration is a key component of hospital antibiotic stewardship interventions. However, its effectiveness in reducing antimicrobial resistance is uncertain and a clear theoretical rationale for the approach is lacking. In this study, we sought to gain a mechanistic understanding of the relation between antibiotic treatment duration and the prevalence of colonisation with antibiotic-resistant bacteria in hospitalised patients.

### Methods and findings

We constructed 3 stochastic mechanistic models that considered both between- and within-host dynamics of susceptible and resistant gram-negative bacteria, to identify circumstances under which shortening antibiotic duration would lead to reduced resistance carriage. In addition, we performed a meta-analysis of antibiotic treatment duration trials, which monitored resistant gram-negative bacteria carriage as an outcome. We searched MEDLINE and EMBASE for randomised controlled trials published from 1 January 2000 to 4 October 2022, which allocated participants to varying durations of systemic antibiotic treatments. Quality assessment was performed using the Cochrane risk-of-bias tool for randomised trials. The meta-analysis was performed using logistic regression. Duration of antibiotic treatment and time from administration of antibiotics to surveillance culture were included as independent variables. Both the mathematical modelling and meta-analysis suggested modest reductions in resistance carriage could be achieved by reducing antibiotic treatment duration. The models showed that shortening duration is most effective at reducing resistance carriage in high compared to low transmission settings. For treated

Research Fellowship (NMRC/Fellowship/0051/ 2017 to MY), the UK Medical Research Council (MR/V028456/1 to BSC), the Wellcome Trust (206736/Z/17/Z to CL and 106698/Z/14/Z to Mahidol Oxford Tropical Medicine Research Programme). The funders had no role in study design, data collection and analysis, decision to publish, or preparation of the manuscript.

**Competing interests:** The authors have declared that no competing interests exist.

individuals, shortening duration is most effective when resistant bacteria grow rapidly under antibiotic selection pressure and decline rapidly when stopping treatment. Importantly, under circumstances whereby administered antibiotics can suppress colonising bacteria, shortening antibiotic treatment may increase the carriage of a particular resistance phenotype. We identified 206 randomised trials, which investigated antibiotic duration. Of these, 5 reported resistant gram-negative bacteria carriage as an outcome and were included in the meta-analysis. The meta-analysis determined that a single additional antibiotic treatment day is associated with a 7% absolute increase in risk of resistance carriage (80% credible interval 3% to 11%). Interpretation of these estimates is limited by the low number of antibiotic duration trials that monitored carriage of resistant gram-negative bacteria, as an outcome, contributing to a large credible interval.

## Conclusions

In this study, we found both theoretical and empirical evidence that reducing antibiotic treatment duration can reduce resistance carriage, though the mechanistic models also highlighted circumstances under which reducing treatment duration can, perversely, increase resistance. Future antibiotic duration trials should monitor antibiotic-resistant bacteria colonisation as an outcome to better inform antibiotic stewardship policies.

---

### Author summary

#### Why was this study done?

- Shortening antibiotic treatment duration is a commonly adopted antibiotic stewardship strategy, with the expectation that it will reduce antimicrobial resistance in treated individuals and in the overall population.

- Antibiotic selective pressure acts predominantly on "bystander" colonising bacteria for resistance, and this depends on the spectrum of coverage, pharmacokinetic and pharmacodynamic properties of individual antibiotics.

- Empirical evidence and an understanding of the mechanisms by which antibiotic treatment duration effects the emergence and spread of antimicrobial resistance are lacking. Understanding the key factors driving the effect of antibiotic treatment duration on resistance carriage will help to inform future research study designs, antimicrobial stewardship interventions, and resource allocation in multimodal control strategies.

#### What did the researchers do and find?

- We modelled within- and between-host dynamics of colonising "bystander" susceptible and resistant bacteria in response to systemic antibiotic treatment and compared the model findings with a systematic review and meta-analysis.

- The meta-analysis found one additional antibiotic treatment day is associated with a 7% absolute increase in risk of resistance carriage when antibiotics administered were not effective against the resistance phenotype in the colonising bacteria.

- For treated individuals, the models showed that shortening antibiotic treatment duration is most effective at reducing resistance carriage when resistant bacteria grow rapidly under antibiotic selection pressure and decline rapidly when stopping treatment.

- At a population level, shortening antibiotic treatment duration is most effective at reducing resistance carriage in high transmission settings.

- Shortening antibiotic treatment duration may increase resistance carriage when the antibiotics administered are effective at eliminating colonising bacteria with a particular resistance phenotype.

## What do these findings mean?

- Shortening antibiotic treatment duration may increase or decrease colonisation by resistant bacteria, dependent upon individual and combined bacterial and antibiotic characteristics.

- The effect of shortening antibiotic treatment duration on colonisation by resistant bacteria colonisation is potentially modest due to short hospitalisation periods and slow decolonisation of resistant bacteria.

- These findings can inform antibiotic stewardship programmes to shorten antibiotic treatment and infection prevention and control policies to reduce transmission of resistant bacteria.

## Introduction

Reducing treatment duration is a common strategy used to reduce antibiotic consumption and a core feature of antibiotic stewardship programmes (ASPs) [1,2]. This includes optimising definitive antibiotic courses for established bacterial infections while ensuring clinical cure, and rapid discontinuation of empiric prescriptions after bacterial infections are ruled out. Compared with restricting prescriptions, this "back-end" approach of minimising treatment duration is deemed safer for patients and more likely to be accepted by physicians [3]. Numerous randomised trials have concluded that short treatment approaches for common bacterial infections are noninferior to long treatment courses in terms of clinical outcomes [4–6].

The primary motivation behind shortening treatment duration is the expectation that it will reduce antibiotic selective pressure for antimicrobial resistance in treated individuals and, over time, will lead to lower prevalence of resistance at a population level [7]. However, while antibiotic selective pressure as a driver of resistance is not in doubt, there are many gaps in our understanding of the relationship between duration and the prevalence of resistance [8–10]. Specifically, clinical trials of reduced antibiotic treatment duration have reported conflicting effects on resistance carriage at both individual and population levels (Table 1).

**Table 1. Details of the antibiotic trials included in the meta-analysis.**

| No. | Indication for antibiotic treatment | Country | Healthcare setting | Age group | Duration of antibiotics (days) | | Antibiotic prescribed | Bacteria detected in surveillance culture | Colonisation site | Duration of follow-up (mean, days) | Total number of participants | Proportion of resistance carriers at the end of treatment or study (resistance carriers/participants who provided samples, %) | | Proportion difference between the long and short arms (long—short, 95% confidence intervals)^ | Reference |
|---|---|---|---|---|---|---|---|---|---|---|---|---|---|---|---|
| | | | | | Short | Long | | | | | | Short arm | Long arm | | |
| 1 | Neonatal late-onset sepsis | European countries | Inpatient | Neonate | 7 | 8 | Meropenem vs standard-of-care | Carbapenem-resistant gram-negative bacteria | Gut | 28 | 272 | 7/94 (7) | 19/101 (19) | 11 (1, 22) | Lutsar, 2020 [37] |
| 2 | Otitis media | USA | Outpatient | Children | 5 | 10 | Amoxicillin/clavulanic acid | Beta-lactamase producing *H. influenzae* | Respiratory tract | 14 | 520 | 28/222 (13) | 38/233 (16) | 4 (−3, 11) | Hoberman, 2016[38] |
| 3 | Uncomplicated urinary tract infection | Turkey | Outpatient | Adults | 1 | 5 | Fosfomycin vs ciprofloxacin | Fosfomycin- or ciprofloxacin-resistant Enterobacteriaceae | Urinary tract | 7 | 260 | 7/77 (9) | 12/65 (18) | 9 (−3, 22) | Ceran, 2010 [39] |
| 4 | Uncomplicated urinary tract infection | the Netherlands | Outpatient | Adults | 3 | 5 | Trimethroprim | Trimethroprim-resistant *E. coli* | Urinary tract | 3 | 324 | 8/66 (12) | 12/63 (19) | 7 (−7, 21) | Merode, 2005[40] |
| 5 | Urinary tract infection in patients with spinal cord injury | Canada | Inpatient | Adults | 3 | 14 | Ciprofloxacin | Fluroquinolone-resistant gram-negative bacteria | Urinary tract | 41 | 60 | 8/30 (27) | 5/30 (17) | −10 (−14, 4) | Dow, 2004 [41] |

^ Difference in proportions were calculated using two-tailed Z-test for 2 proportions.

Antibiotics promote resistance by killing or inhibiting the growth of susceptible microbes while allowing resistant ones to grow. This selective pressure for resistance most clearly applies to pathogens directly targeted by the antibiotic when used to treat or prevent infections. Examples of pathogens that may become resistant during treatment include *Mycobacterium tuberculosis*, fluoroquinolone-resistant Enterobacterales, and human immunodeficiency virus, especially with inappropriate dosing and duration [11,12]. More recently, it has been recognised that, for most bacterial pathogens, antibiotic selection pressure largely results from "bystander" selection, which occurs when colonising bacteria are exposed to antibiotics that are not intentionally targeting them [13]. The impact of bystander exposures to antibiotics depends on their spectrum of coverage, pharmacokinetic and pharmacodynamic properties, and bioavailability where the bacteria are located. Antibiotic resistance in colonising bacteria is clinically important because asymptomatic carriage of these bacteria typically precedes invasive infection, and preventing colonisation will reduce associated morbidity and mortality [14–16]. For these reasons, this study focuses on the "bystander" selective pressure resulting from different durations of antibiotic treatment on colonising bacteria.

Emergence of resistance in colonising bacteria may be due to selection of resistant phenotypes that are present at the start of treatment but at low abundance or acquired by transmission during treatment, as a result of horizontal gene transfer, or as a result of de novo mutations. The effect of treatment duration on development of resistance will depend on the parameters governing these processes and their relative importance.

To gain a mechanistic understanding of this relation between antibiotic treatment duration and the prevalence of colonisation with antibiotic-resistant bacteria in hospitalised patients, we modelled within- and between-host dynamics of susceptible and resistant bacteria in response to systemic antibiotic treatment. We then sought to compare the model findings with empirical observations by performing a systematic review and meta-analysis.

## Methods and materials

### Modelling dynamics of colonising bacteria in response to antibiotic treatment

Three stochastic agent-based models with increasing complexity and biological realism were constructed. From here on, they will be referred to as simple exclusive colonisation model, co-colonisation model, and within-host growth model (Fig 1).

In all models, we simulated individual patients (agents) in a single hospital ward environment. The patients' colonisation status could change due to (i) differential growth/killing rates of resistant and sensitive bacterial populations within a patient; (ii) transmission events between patients; and (iii) loss of resistance carriage (Fig 1). Because the population size under consideration is small and local fade-out events (when resistant populations reach zero) are potentially important, we considered stochastic implementations of these models (Fig 2 and Section S1.1 in S1 Text), i.e., changes in colonisation status took place with a probability randomly drawn from a distribution.

Within the same ward, antibiotics (administered via both intravenous and oral routes) were prescribed with the same mean duration on or during admission. Exact duration for each antibiotic course was drawn from a uniform distribution (range for short duration 3 to 7 days, range for long duration 14 to 21 days). We considered 2 scenarios. In the first, the resistant phenotype was assumed to be susceptible to one of the administered antibiotics, e.g., carbapenems against third-generation cephalosporin resistance. In the second scenario, there was no effective antibiotic available against the resistant phenotype in the colonising bacteria, e.g., colistin or carbapenem resistance. Subsequently, we refer to these 2 scenarios as "administered

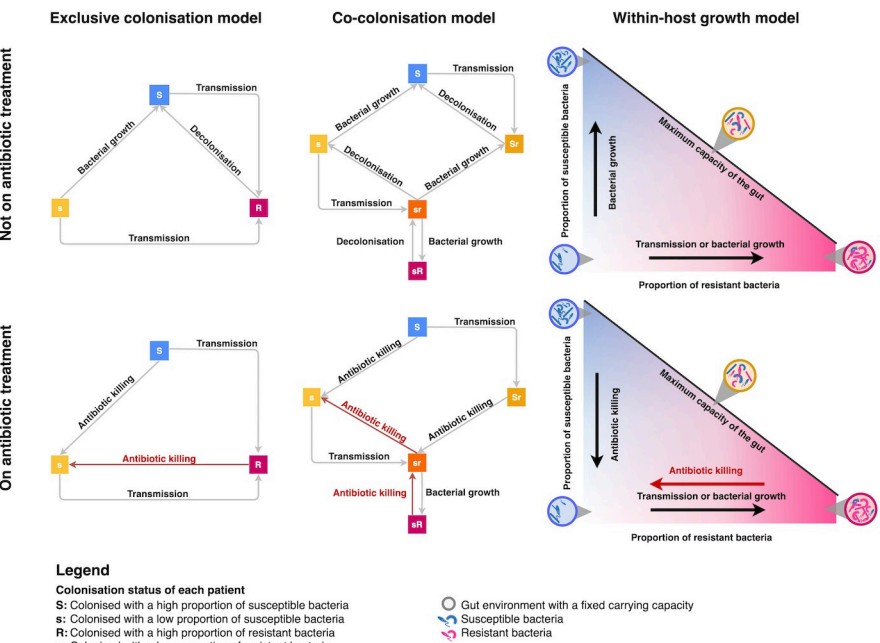

**Fig 1. Host states and transitions in the exclusive colonisation, co-colonisation, and within-host growth models.**
These flow diagrams describe within-host changes in resistant and sensitive bacteria carried by each individual patient in response to antibiotic consumption, bacterial growth, and transmission events. In the 3 models, all patients were assumed to be always colonised with bacteria. In the exclusive colonisation model, patients can be colonised with resistant bacteria (R), and high or low levels of sensitive bacteria (S or s), but not both resistant and sensitive at the same time due to complete bacterial interference. The co-colonisation model makes a more realistic assumption that patients may be colonised with both resistant and sensitive bacteria at the same time with different levels of abundance. Similarly, in the within-host growth model, patients carry both resistant and sensitive bacteria but their combined populations cannot exceed a maximum carrying capacity. Blue, orange, and pink squares/circles represent the host states when the host is carrying susceptible, a mix of susceptible and resistant, and resistant bacteria, respectively. The red arrows highlight antibiotic killing of resistant bacteria. All transmission and decolonisation events apply to resistant bacteria.

antibiotics have activity against resistant and susceptible organisms" and "administered antibiotics have activity only against susceptible organisms."

The models assumed that antibiotics act within-host by selectively promoting the growth of existing resistant bacteria in the gut [17,18], and between-host by predisposing the treated individual to become more likely to transmit resistant bacteria as a consequence of a higher bacterial load (Fig 1) [19]. Spontaneous decolonisation of resistant bacteria was assumed to only occur when a patient was not receiving antibiotics to which the colonising bacteria were resistant.

In the exclusive colonisation model, patients could be carriers of either susceptible or resistant bacteria but not of both at the same time [20]. In this model, for a carrier of susceptible bacteria to become a resistance carrier required transmission of resistant bacteria from another carrier in the ward, i.e., we did not consider de novo resistance emergence or horizontal transfer from other components of the flora. In the other 2 more complex models, patients could carry both susceptible and resistant bacteria simultaneously. In the co-colonisation model, these abundances were dichotomised into high and low, and only bacteria carried at a high level were able to be transmitted to other patients [21]. In the within-host growth model, bacterial population sizes could vary continuously but had to be above a threshold in order to be capable of transmission to others [22]. We also assumed a total carrying capacity for gram-negative bacteria [23] and note that newly admitted patients may carry fewer bacteria than the

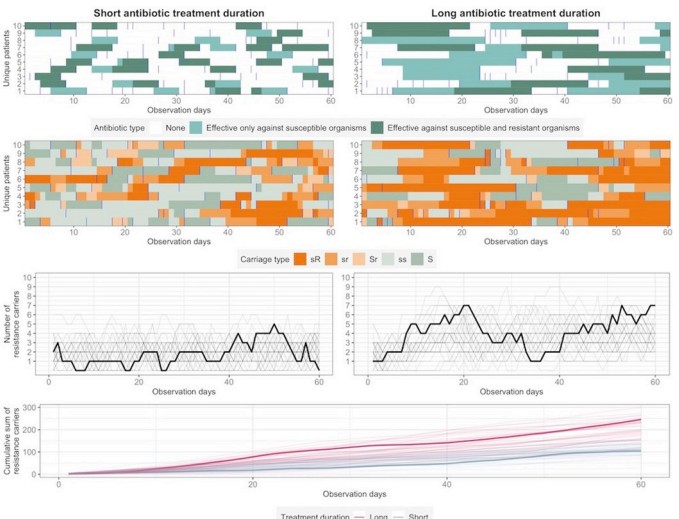

**Fig 2. Example output from the co-colonisation model.** This example simulation shows two 10-bedded wards with the same transmission risk of resistant bacteria, patient lengths of stay, proportion of patients who required antibiotic treatment, and proportion of patients who were resistance carriers upon admission. The left column panels represent the ward where antibiotic durations were short (mean of 5 days), while the right column panels represent the ward where antibiotic durations were long (mean of 15 days). Top row panels: Antibiotic treatment types and duration for one single iteration. Each square in the graphs represents one patient day. Light and dark green squares indicate one day of antibiotic effective only against susceptible organisms or both susceptible and resistant organisms. Vertical blue lines represent new admissions to the ward. Second row panels: The carriage status for each patient on a particular day for one single iteration. The increasingly darker shades of orange and green indicate an increasing amount of resistant and susceptible colonising bacteria carried by each patient, respectively. Third row panels: The number of antimicrobial resistance carriers in each ward per day for 40 iterations. Fourth row panel: The cumulative number of resistance carriers in each ward over 60 observation days for 40 iterations. In the third and fourth row panels, each line represents output from one iteration. The thick lines are outputs from the single example iteration illustrated in the top 2 panels.

carrying capacity as we would expect some patients were prescribed antibiotics prior to admission. The mean total carrying capacity for each iteration was drawn from a lognormal distribution (Table A in S1 Text). In the latter 2 models, an individual could become a resistance carrier from within-host selection of preexisting resistant bacteria through antibiotic consumption.

While the modelling framework we have adopted is quite general, here we focus on parameter space that are appropriate for gram-negative bacteria as they are the commonest cause of urinary tract and bloodstream infections and frequently associated with multidrug-resistant hospital-acquired infections. The predominant mechanisms for resistance dissemination in gram-negative bacteria colonising the gut are horizontal transfer of mobile genetic elements and clonal expansion [24].

We first explored the models by varying pairs of parameters over a grid of values while holding the other parameter values constant. This was followed by global sensitivity analysis using Latin Hypercube sampling and Partial Rank Correlation Coefficient (S1.3 Section in S1 Text). Real-world parameter values were obtained from the literature and used in these explorations (Table A in S1 Text). For those parameter values not found in the literature, we explored the largest reasonable range of parameter values. In all results shown below, each simulation was produced from at least 50,000 iterations (100 repeats of 500 unique sets of parameters selected within the ranges shown in Table A in S1 Text) over 300 days. The computer code is publicly available (https://github.com/moyinNUHS/abxduration_abm).

The models were used to assess how changing duration of antibiotic treatment affected the risk of resistance colonisation at both individual and population levels. Three types of outcomes were considered: resistance carriage among (i) treated patients and, therefore, directly affected by different treatment durations; (ii) overall resistance carriage within a ward population that consisted of treated and untreated patients, i.e., indirectly affected by treatment duration received by those treated patients; and (iii) patients who were not carriers of resistant bacteria when admitted to the ward. To evaluate reducing antibiotic duration as an intervention, all the model outputs were assessed as absolute difference between the short and long wards.

All simulations were performed with R version 4.0.4 (2021-02-15) [25] using packages msm [26], pse [27], and spartan [28]. Code review was done using unit tests for each function's base scenarios, which can be found under /unit_tests in the source code.

## Systematic review

To compare model findings with empirical evidence, we performed a systematic review of antibiotic duration randomised controlled trials, which reported resistance carriage. We searched MEDLINE and EMBASE for randomised controlled trials published from 1 January 2000 up to 4 October 2022, which allocated participants to varying durations of systemic antibiotic treatments. Search strings are provided in Table A in S2 Text. Studies that compared antibiotics to no antibiotic treatment were excluded. Quality assessment of the studies was performed using the Cochrane risk-of-bias tool for randomised trials (RoB 2 tool) [29]. MY and MO reviewed the shortlisted articles, extracted and verified the underlying data for the meta-analysis independently. This study is reported as per the Preferred Reporting Items for Systematic Reviews and Meta-Analyses (PRISMA) guideline (S1 Prisma Checklist). The funders had no role in the design and conduct of the review. The systematic review reviewers (MY and MO) have no competing interest.

The study methodologies and outcomes were expected to be too diverse to be meaningfully included in a single meta-analysis. Instead, we prespecified a relatively homogeneous group defined by studies that (i) measured asymptomatic carriage of resistant bacteria as an outcome; and (ii) collected cultures at predefined time point(s) for surveillance (not clinically indicated) during follow-up. The specific inclusion criteria and search strategy can be found in Section S2.1 in S2 Text and Table A in S2 Text. The indications for antibiotic treatment in the trials included treatment or prevention of bacterial infections, decolonisation of resistant bacteria, and anti-inflammation for autoimmune diseases. The sites of colonisation included digestive, respiratory, and urinary tracts. We considered resistance carriage when the antibiotic prescribed in the trial was not effective against the specific type of resistance, e.g., ciprofloxacin was prescribed in a trial that monitored carriage of fluoroquinolone-resistant bacteria. No trials monitored resistance phenotypes, which the prescribed antibiotic was effective against.

The specific aims of the systematic review were to (i) evaluate the quality of evidence in the current literature of antibiotic duration effects on resistance carriage; (ii) describe and summarise the findings from these studies; and (iii) compare the findings to the model outputs.

## Meta-analysis

From the studies identified in the systemic review, we selected those that monitored resistant gram-negative bacteria carriage for a meta-analysis. The meta-regression analysis was performed using a Bayesian model to estimate the change in the daily risk of acquiring resistant bacteria colonisation per day of antibiotic consumption. The dependent variable in the meta-regression analysis was the number of patients colonised with resistant bacteria in a given arm

of a given trial, together with the associated denominator. These data were analysed with a logistic regression using antibiotic duration administered to the patients in each arm and elapsed time from antibiotic administration to surveillance culture as independent variables. The regression model assumed each arm in each trial to be independent (fixed effect). Details of the models can be found in S2.4 Section in S2 Text.

We also considered 2 other models. In the first alternative model, we included the healthcare setting as an additional independent variable. In the second, we allowed slopes and intercepts to vary between trials (random effect). Model comparison was done using the Widely Applicable Information Criterion (WAIC), where lower values indicate improved model fit [30].

We implemented the meta-regression models in JAGS using the R2jags package [31] and performed all analysis in R version 4.0.4 (2021-02-15) [25]. All analysis codes are available at https://github.com/moyinNUHS/abxduration_abm.

## Results

### Individual and population resistance dynamics with antibiotic treatment

We first consider as an illustrative example of model behaviour a ward where 25% of the patients carried resistant bacteria on admission and half of the patients were given antibiotics upon admission (Fig 3). In the scenario where the patients were given an antibiotic with activity against both susceptible and resistant colonising bacteria (Fig 3, top 2 panels), in all 3 models, carriage of the resistant bacteria declined both in treated individuals and at the population level with longer antibiotic treatment duration.

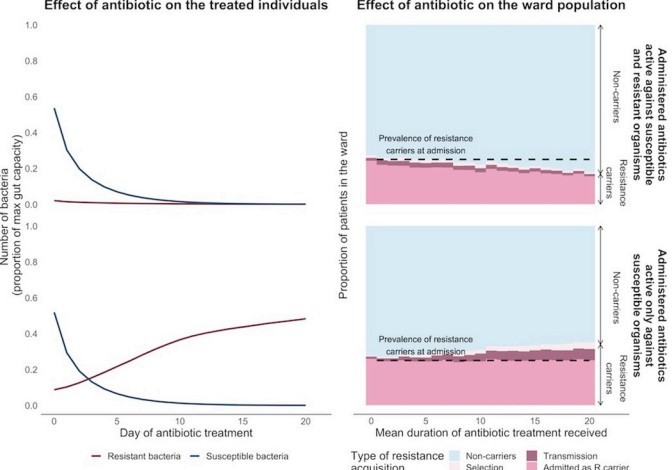

**Fig 3. Effect of antibiotic duration on antimicrobial resistance at individual and population levels (results shown are from the within-host growth model, but qualitatively similar results are obtained with the other 2 models).** A ward of 50 patients was simulated where 25% of the patients were already colonised with resistant bacteria when admitted to the ward. Half of these 50 patients were administered antibiotics assumed to start as soon as patients were admitted. Left-sided panels depict the mean within-host dynamics of susceptible (blue line) and resistant (pink line) bacteria with increasing days of antibiotic treatment among the treated patients who received 20 days of antibiotics. Right-sided panels show the equilibrium prevalence of resistance carriers on the ward as a function of antibiotic treatment durations. The proportion of resistance carriers (represented by shades of pink) and noncarriers (represented by blue) in the ward are plotted over antibiotic treatment time. The gradient of the pink-shaded slopes indicates the effect of antibiotic duration on resistance carriers in the wards. Top panels show the scenario where administered antibiotics have activity against both susceptible and resistant bacteria. Bottom panels show the scenario where administered antibiotics have activity only against susceptible organisms.

When antibiotics had activity only against susceptible organisms (Fig 3, bottom 2 panels), the "bystander" selection on resistant bacteria led to an increased prevalence of resistance carriers with longer treatment. More resistance carriers in the ward then acted as a reservoir to further spread resistance to noncarriers.

## Key parameters in the relationship between antibiotic duration and resistance carriage

The global sensitivity analysis identified transmission rates and prevalence of resistance on admission as important effect modifiers of the relationship between treatment duration and resistance prevalence (Fig 4, "baseline carriage status" and "resistance transmission" rows). Reducing treatment duration was more effective in reducing resistance carriage in the overall ward population when the transmission rate was high regardless of whether the administered antibiotics were active against resistant organisms.

Conversely, the lower the resistance prevalence on admission, the more effective reducing treatment duration was in reducing resistance carriage at a population level. This is because importation of resistance carriers is not affected by antibiotic treatment. Hence, for the ward population, the lower the prevalence of resistance carriage on admission, the greater effect antibiotic duration has on overall resistance.

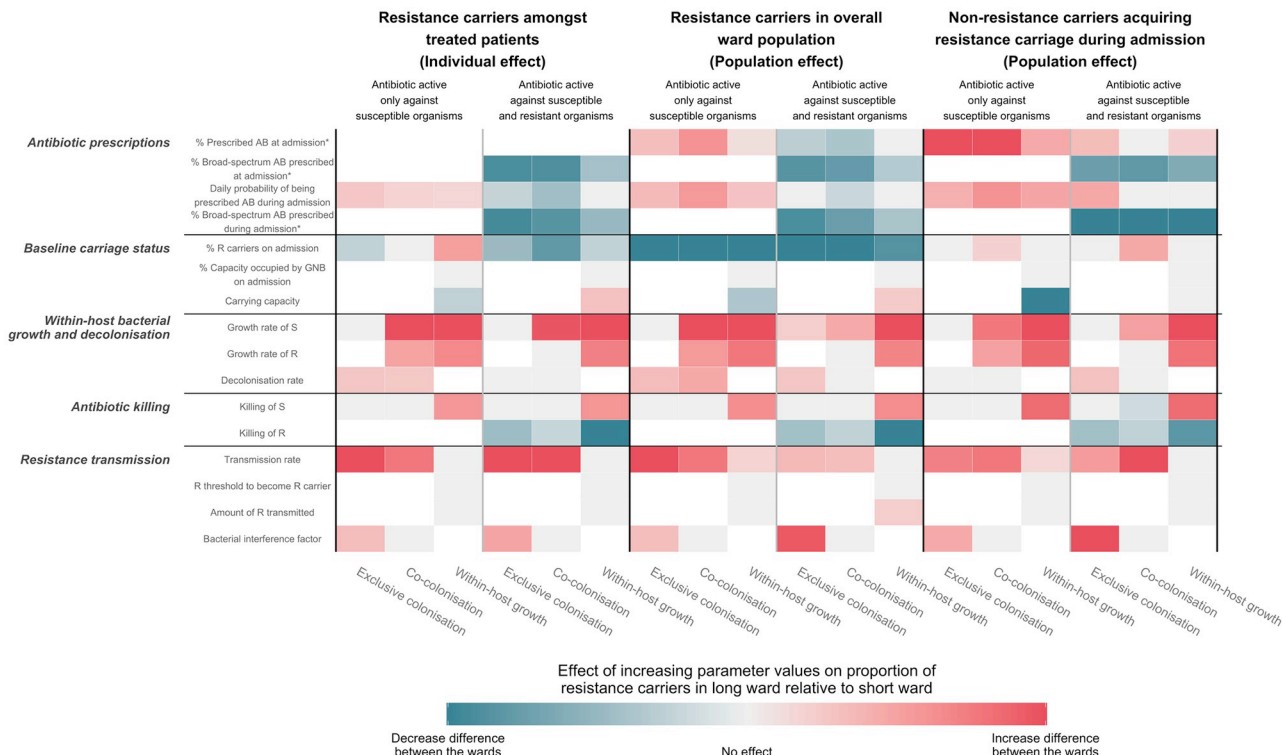

**Fig 4. Global sensitivity analysis.** Partial rank correlation coefficients for the various parameters are shown in the heatmap. Parameters from the 3 different models are grouped according to the main within- and between-host processes they describe. The 3 panels (separated by black vertical lines) represent the 3 types of outcomes assessed. Within each panel, the left 3 columns refer to the scenario where administered antibiotics have activity only against susceptible organisms; the right 3 columns refer to the scenario where administered antibiotics have activity against resistant and susceptible organisms. Pink-, teal-, and grey-coloured rectangles indicate that as each parameter value increased, the difference in high-abundance resistance carriers between the long- and short-duration wards increased, decreased, and did not change, respectively. AB, antibiotic; GNB, gram-negative bacteria; R, Resistant; S, Susceptible. White rectangles indicate that the parameter was not found in the model or not applicable. *Parameters for the percentage of patients who were administered antibiotics are not relevant for the outcome observed in the treated patients and therefore coloured white.

However, among patients who were treated and who were noncarriers on admission, reducing treatment duration tended to be more effective when the prevalence of resistance carriage on admission was high. This is because in these subsets of patients, the gain in resistance carriage was mainly through transmission events from existing carriers rather than within-host bystander selection.

Within-host, a rapid growth rate of susceptible bacteria after stopping antibiotics and resistant bacteria while receiving antibiotic treatment resulted in more resistance carriers in the ward administered longer treatment duration compared to the ward administered short duration (Fig 4, "within-host bacterial growth and decolonisation" row).

## Within-host dynamics of colonising bacteria with varying antibiotic treatment duration

Bacterial growth and decolonisation rates in the absence of antibiotics were important within-host factors in the relationship between treatment duration and resistance carriage regardless of whether administered antibiotics were active against resistant organisms. When resistance phenotypes are resistant to antibiotic treatment, under both fast and slow growth scenarios, we find that long duration treatment allows the resistant population to become established at a much larger proportion of the total bacterial load at the end of treatment, which in turn results in a longer time for the resistant population to decline to low levels once treatment ceases (Fig 5).

During antibiotic treatment, susceptible bacteria declined while resistant bacteria were able to grow. Hence, rapid resistant bacteria growth enabled hosts to transmit resistance onward to others in the ward (Fig 5, solid pink lines). Upon termination of antibiotics, rapid susceptible bacteria recovery suppressed resistant bacteria growth (Fig 5, solid blue lines). However, when susceptible bacteria recovered slowly after termination of antibiotics, patients who were administered antibiotics could remain colonised with resistant bacteria for prolonged periods regardless of the treatment duration (Fig 5, dashed pink lines).

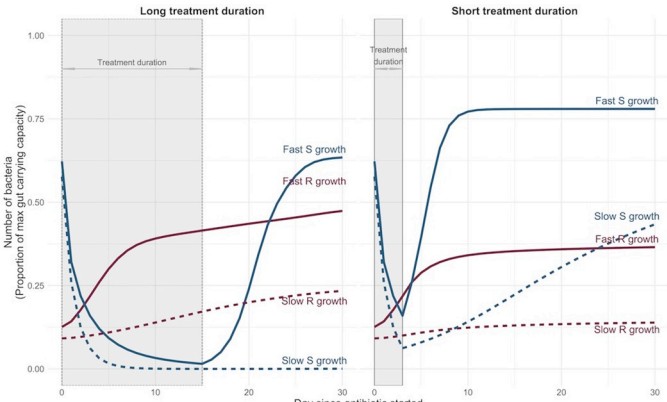

**Fig 5. Example output from the within-host growth model showing effect of antibiotic duration on competitive growth of susceptible (blue) and resistant (pink) colonising bacteria at different growth rates.** These graphs illustrate an example where the susceptible and resistant bacteria within the same host were exposed to 15 days (left panel) versus 3 days (right panel) of antibiotic treatment. This antibiotic was capable of killing susceptible bacteria but ineffective against resistant bacteria. All parameter values other than bacterial growth rates were kept the same. Rapid bacterial growth is represented by solid lines. Slow bacterial growth is represented by dashed lines.

## Resistance transmission and importation on the effects of antibiotic duration

In the scenario where administered antibiotics have activity only against susceptible organisms, antibiotic treatment had separate effects on the treated nonresistance carriers and resistance carriers (Fig 6). For the noncarriers, antibiotics increased the risk of acquiring resistant bacteria by reducing the abundance of colonising bacteria to below the carrying capacity. For the existing carriers, antibiotics preferentially promoted the growth of resistant bacteria, increasing these patients' probability of transmitting resistance to others. Hence, higher transmission rates amplified the effect of treatment duration on increasing resistance carriers by bridging the transfer of resistance from the carriers to the at-risk noncarriers.

When administered, antibiotics were active against the resistant organisms; there remained areas of parameter space where longer treatment duration led to more resistance (Fig 6, Panel B top row). This was observed at high resistance importation and transmission rates, especially for treated patients and patients who were noncarriers at admission. This is because in these patients, longer antibiotic treatment reduced the abundance of colonising bacteria and increased these patients' risk of acquiring resistance.

## Overall effect of antibiotic duration on resistance carriage

The models showed generally modest and highly heterogenous effects of antibiotic duration on resistance carriage (Fig 7). When antibiotics have activity only against susceptible organisms, longer antibiotic duration resulted in higher prevalence of resistance carriage (Fig 7, right-sided panels). In contrast, when antibiotics have activity against both resistant and susceptible organisms, longer antibiotic duration resulted in either an increase or decrease in the prevalence of resistance carriers (Fig 7, left-sided panels).

## Evidence from randomised controlled trials

Initial search of the MEDLINE and EMBASE databases returned 2,649 unique publications. Out of these 206 were randomised trials that compared antibiotic treatment durations. Ten of these trials collected surveillance cultures for colonising bacteria during follow-up visits and were included in the qualitative synthesis. A meta-regression analysis was performed using 5 out of the 10 trials (Table 1). One was excluded because the antibiotic duration in the long arm and the follow-up time were very prolonged compared to the other studies (104 days treatment compared to 5 to 14 days, and 365 days follow-up compared to 3 to 41 days) [32]. One was excluded because the trial reported a mixture of antibiotics and multidrug-resistant bacteria carriage without specifying if the organisms were gram-positive or gram-negative [33]. Three others were excluded because the study only collected gram-positive bacteria in surveillance cultures [34–36]. The PRISMA diagram can be found in the Fig A in S2 Text. Complete data extracted from these randomised trials can be found at https://github.com/moyinNUHS/abxduration_abm.

Among the 10 trials, the indications for antibiotic treatments in 9 were for treatment of infections (urinary tract infections (3), otitis media (1), sepsis (1), acute respiratory illness (4)), and in one was for *Pseudomonas* spp. in noncystic fibrosis bronchiectasis. Six were performed in outpatient settings with mean follow-up periods ranging from 3 to 365 days; 4 were in hospital settings with mean follow-up periods ranging 28 and 90 days. Five trials enrolled only adults, while the other 5 enrolled only children. All were performed in high- or upper-middle-income countries. Antibiotic durations in the short arms ranged from 3 to 14 days (median 5 days), while in the long arms, the durations ranged from 5 to 104 days (median 10 days).

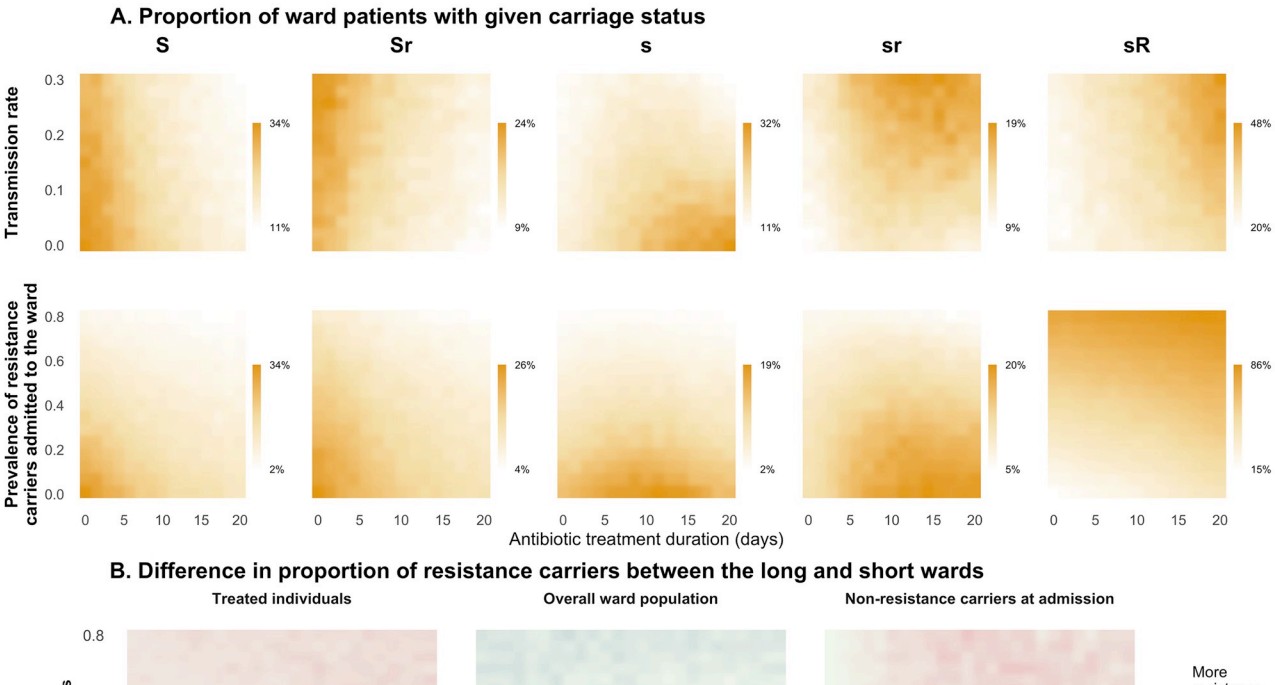

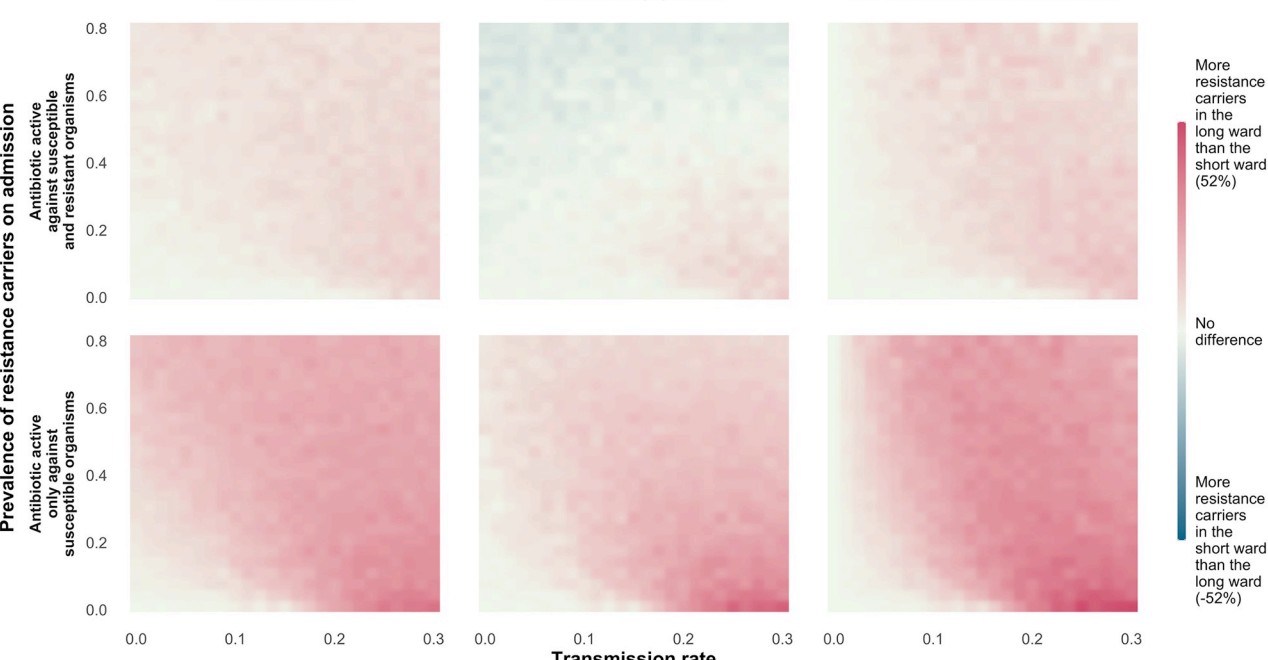

**Fig 6. Effects of transmission rate and baseline prevalence of resistance carriers at admission on resistant bacteria carriage in the scenario where administered antibiotics have activity only against susceptible organisms.** The values shown (represented by the coloured pixels) were derived from the models when they have reached equilibrium. Panel A: Output from co-colonisation model under various parameter values (y-axis) for (i) transmission rate and (ii) prevalence of resistance carriers admitted to the ward with increasing antibiotic duration (x-axis). The coloured pixels indicate the proportion of ward patients with a given carriage status where *S/s* represent high and low proportions of susceptible bacteria and *R/r* represents high and low proportions of resistant bacteria. Increasing antibiotic duration increased the risk of the treated patients acquiring resistance carriage by suppressing the susceptible bacteria population. This resulted in shifts with increasing duration of treatment from *S* and *Sr* states to predominantly state *s* at low transmission rates and to predominantly state *sr* and *sR* at higher transmission rates. Panel B: Output from the exclusive colonisation model. (i) Transmission rate (x-axis) and (ii) prevalence of resistance carriers admitted to the ward (y-axis) were varied while other parameters were fixed to explore the difference in proportion of resistance carriers between the wards administered long and short antibiotic duration (coloured pixels). Red, white, and blue pixels indicate that there were more resistance carriers in the long ward (i.e., shortening treatment duration was effective at reducing resistance carriers), no difference between the short and long ward (i.e., shortening treatment duration had no effect on resistance carriers), and more resistance carriers in the short ward (i.e., prolonged treatment duration was effective at reducing resistance carriers), respectively. The top and bottom rows describe the scenarios where administered antibiotics have activity against both susceptible and resistant organisms and only susceptible organisms, respectively. The columns represent the 3 populations in which the outcomes were assessed: (i) treated individuals; (ii) overall ward population; and (iii) noncarriers on admission.

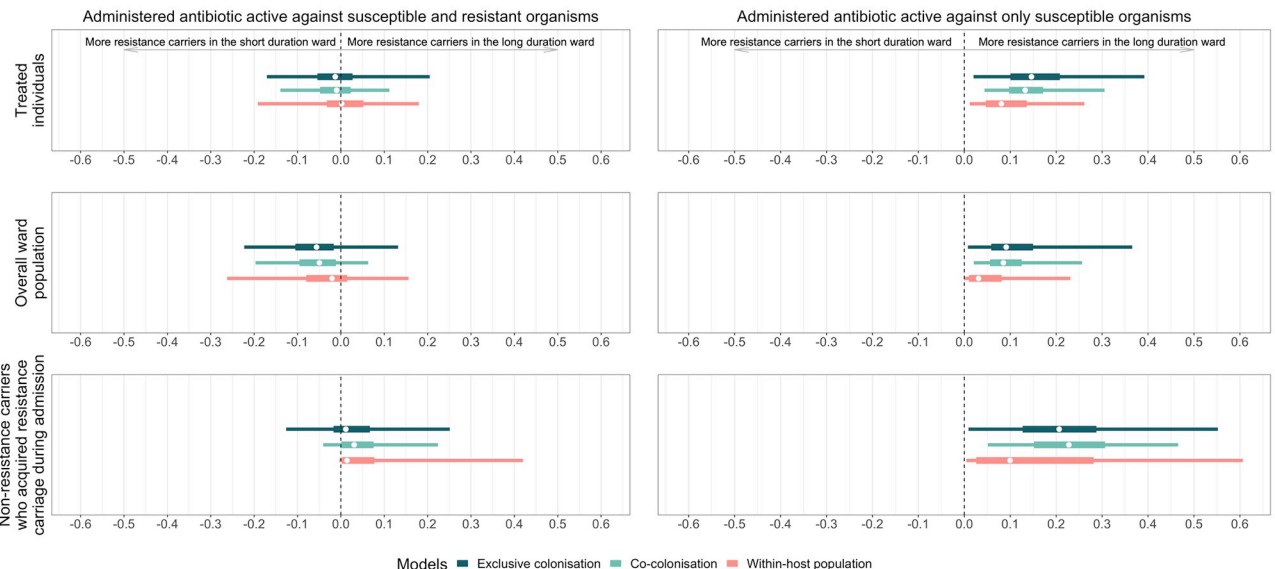

**Fig 7. The difference in proportion of resistance carriers between the wards administered long and short antibiotic duration (x-axis) under different antibiotic treatments.** The 3 panel rows indicate the outcomes assessed: (i) treated individuals; (ii) overall ward population; and (iii) noncarriers on admission. A total of 500 parameter combinations were used in the simulations sampled from uniform distributions (Table A in S1 Text). For each sampled parameter combination, we averaged 100 iterations of the stochastic model. Each coloured bar represents the average output distribution for each set of parameter values. In each bar, the white dots represent the median and the thick and thin bars represent the 50% and 95% interval ranges.

The meta-analysis found an odds ratio of acquiring gram-negative resistance carriage with one additional day of antibiotics of 1.08 (80% credible interval 1.04% to 1.12%). This translates to an absolute 7% increase in daily probability of acquiring resistance carriage given a baseline daily probability of 0.1 (Fig 8). Sensitivity analyses using data only from studies with surveillance cultures collected up to 30 days after antibiotic treatment and using different priors produced similar results (Table C in S2 Text).

Three out of 5 studies were deemed to be at risk of bias due to the high proportion of participants who were lost to follow-up and did not provide surveillance samples (Table D in S2 Text). However, all studies were assessed to be at low risk of bias due to deviation from intended interventions.

## Discussion

The 3 models reached broadly similar conclusions: Among treated individuals, shortening antibiotic duration is most effective at reducing resistance carriage when the bacterial dynamics respond rapidly to antibiotic treatment. When stopping antibiotics leads to rapid decline in the resistant strain's abundance, shortening duration is more likely to result in a substantial reduction in the prevalence of resistance [18]. At a population level, high transmission of resistant bacteria between hosts was a key factor in the efficacy of shortening treatment duration at reducing resistance carriage.

Longer antibiotic treatment may decrease or increase the prevalence of resistance carriers, depending (among other factors) on the availability of effective antibiotics against particular resistance phenotypes. The meta-analysis, using data from randomised controlled trials, highlighted the increased incidence of resistance colonisation with treatment duration when there was no effective antibiotic treatment available. The models also showed that even when antibiotics administered are active against a resistance phenotype, longer duration may

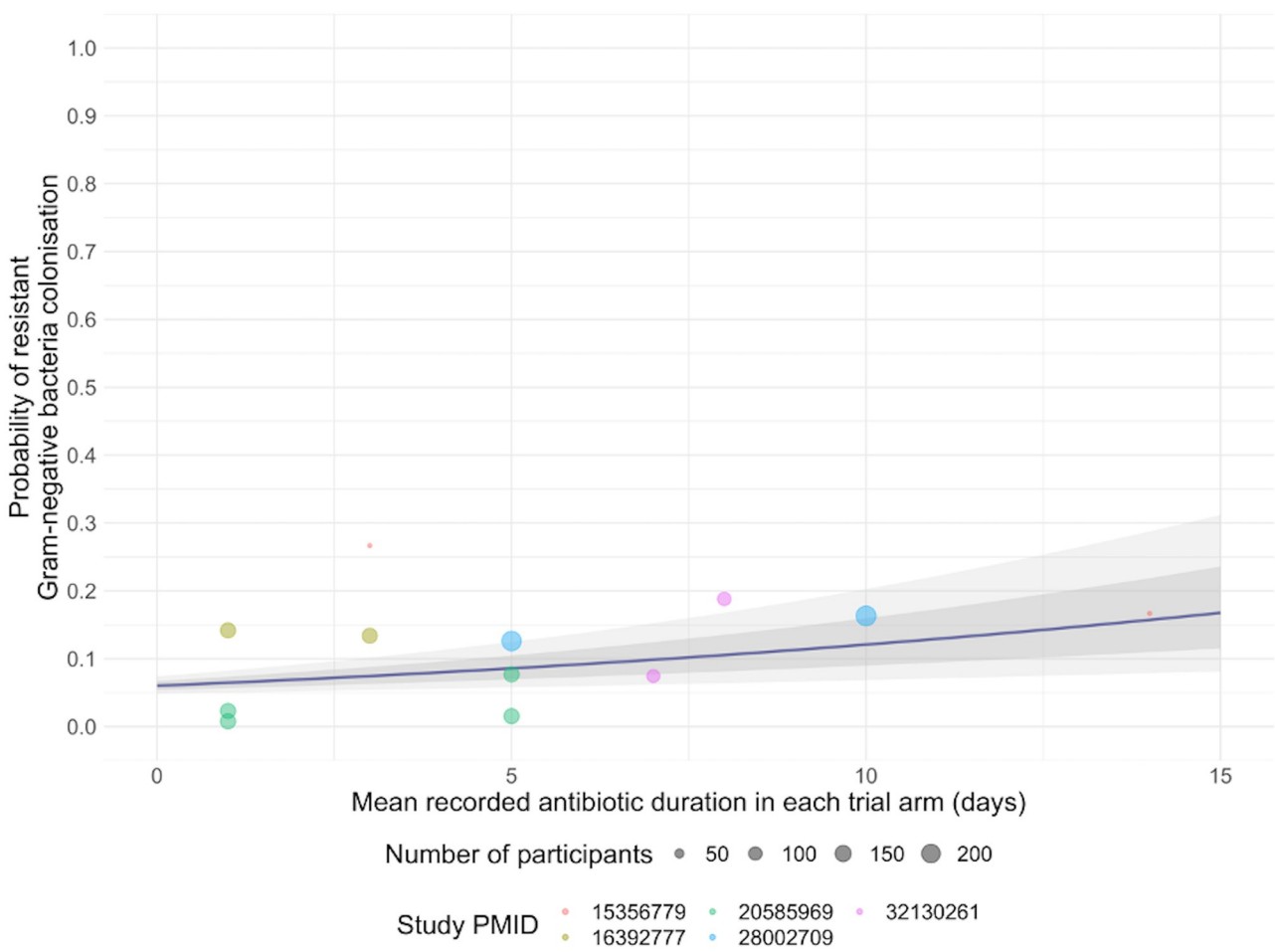

**Fig 8. Daily risks of colonisation by antibiotic-resistant gram-negative bacteria given days of antibiotics prescribed reported by the randomised controlled trials included in the meta-analysis.** Daily probability of colonisation by antibiotic-resistant gram-negative bacteria (y-axis) is shown against mean duration of antibiotics (x-axis) reported in each trial. Each colour represents one trial. Each bubble represents a single arm in one trial, where the diameter of the bubble corresponds to the number of participants for the arm in the trial. The analysis allowed the relationship between colonisation and antibiotic duration to vary across each arm in the trials; the blue line corresponds to the mean relationship between the 2 considering all included trials. The light grey shaded areas are the associated 80% and 50% credible intervals.

increase resistance carriers especially among the treated individuals and noncarriers when the prevalence of resistance carriers on admission and transmission rates are high.

We searched MEDLINE and EMBASE for modelling studies that evaluated antibiotic treatment duration on antimicrobial resistance and found 3 related publications. The first study by D'Agata and colleagues is a theoretical agent-based model, which concluded that rapid initiation of treatment and minimising duration were likely to reduce antimicrobial resistance in a hospital epidemic setting [42]. The second study by D'Agata and colleagues focused on within-host dynamics of resistance by effective and ineffective antibiotic killing [43]. It found that shorter and early interruption of antibiotic therapy selected resistant strains when antibiotic killing was effective. The other study is by Blanquart and colleagues, in which antibiotic prescription and antimicrobial resistance data specific to *Streptococcus pneumoniae* from the Israeli community were used [44]. This study suggested that reducing the number of courses of antibiotics might be a more efficient strategy for reducing antimicrobial resistance than reducing treatment duration. There were no studies that directly addressed the relationship

between antibiotic duration and antimicrobial resistance in both individual and population levels specific to healthcare settings.

We did not find any cluster-randomised trials of antibiotic duration that monitored resistance colonisation at the population level. Such trials would have allowed us to compare the individual versus population effects from antibiotic selection. Since most resistance selection in common bacterial pathogens is due to "bystander" selection [45], trials that only look at resistance in bacteria that are causing infections being treated are potentially missing a big part of the picture. Future trials should collect surveillance cultures to evaluate the effect of duration on resistance carriage.

The modest effect of shortening antibiotic duration on reducing resistant bacteria colonisation can be partially explained by the unique patient and antibiotic prescribing characteristics in healthcare settings. These include typically short lengths of stay and a slow rate of decolonisation of resistant bacteria in those previously exposed to antibiotics [17,18]. Admitted patients under antibiotic treatment acquiring resistant bacteria carriage can thus remain as carriers during relatively brief hospitalisation periods even after antibiotic treatment is stopped. This initial rapid increase in risk of acquiring resistance bacteria carriage during the first few days of antibiotic treatment could reduce the effectiveness of shortening treatment duration at reducing resistance.

Our findings highlight an important interaction between shortening antibiotic treatment and reducing transmission of resistant bacteria through infection prevention and control. This is intuitive because in addition to selecting for within-host resistance, antibiotic use also increases the risk of colonisation with resistant bacteria and subsequent prolonged colonisation [17,18]. Shortening treatment duration has the potential to reduce a patient's risk of acquiring resistant bacteria, while infection prevention and control measures reduce transmission of resistant bacteria to these at-risk patients.

There are important caveats and limitations in our study. Firstly, despite performing a comprehensive systematic review, some parameter values were not available in the literature or were taken from animal or in vitro experiments. In such cases, we explored the largest reasonable range of parameter values. Secondly, emergence and spread of antimicrobial resistance under antibiotic selective pressure are complex and cannot be fully described with any of the models presented. It is reassuring, however, that all 3 versions of the models produced similar conclusions. Thirdly, for the meta-analysis, there were few antibiotic duration trials that monitored resistance carriage as an outcome. This contributed to large credible intervals in the daily resistance colonisation risk with antibiotic treatment. These limitations highlight important gaps in existing literature and the conduct of antibiotic duration trials. Fourthly, our models did not account for de novo mutations during treatment, which are an important source of resistance for certain organisms such as *Mycobacterium tuberculosis* [46]. Instead, we focused on pathogens in which colonisation with resistant phenotypes is primarily acquired through transmission and horizontal gene transfer. These are highly clinically relevant gram-negative bacteria such as carbapenemase-producing Enterobacterales, extended spectrum beta-lactamase-producing Enterobacterales, and carbapenem-resistant *Acinetobacter* spp. Lastly, the agent-based models captured both individual and population-level effects of antibiotic selection pressure. However, the trials included in the meta-analysis were individually randomised, and results therefore reflected only the direct individual antibiotic selection effects.

Understanding the key factors driving the effect antibiotic duration on resistance carriage will inform future research study designs, antimicrobial stewardship interventions, and resource allocation in the overall control strategies. The practical implications from our findings are that interventions for shortening antibiotic treatment duration are potentially most

effective when antibiotics are stopped as early as feasible without compromising treatment success for the target pathogen, especially in high transmission settings.

## Supporting information

**S1 Text. Supporting information: Model details and exploration.** Table A. Parameters used in the models and their respective ranges obtained from literature review.
(DOCX)

**S2 Text. Supporting information 2: Systematic review methodology.** Table A. Search terms used in the systematic review. Table B. Model comparisons. Table C. Sensitivity analysis. Table D. Quality assessment of the randomised controlled trials included in the meta-analysis.
(DOCX)

**S1 PRISMA Checklist. PRISMA checklist.**
(DOCX)

## Acknowledgments

We would like to thank Anastasia Hernandez-Koutoucheva, Ricardo Sempedro, and Jiraboon Tosanguan for performing code review.

## Author Contributions

**Conceptualization:** Yin Mo, Mathupanee Oonsivilai, Cherry Lim, Rene Niehus, Ben S. Cooper.

**Data curation:** Yin Mo, Mathupanee Oonsivilai, Ben S. Cooper.

**Formal analysis:** Yin Mo, Mathupanee Oonsivilai, Ben S. Cooper.

**Funding acquisition:** Yin Mo, Ben S. Cooper.

**Investigation:** Yin Mo, Mathupanee Oonsivilai, Cherry Lim, Rene Niehus, Ben S. Cooper.

**Methodology:** Yin Mo, Mathupanee Oonsivilai, Ben S. Cooper.

**Project administration:** Yin Mo, Ben S. Cooper.

**Resources:** Yin Mo, Ben S. Cooper.

**Software:** Yin Mo, Mathupanee Oonsivilai.

**Supervision:** Ben S. Cooper.

**Validation:** Yin Mo, Mathupanee Oonsivilai, Ben S. Cooper.

**Visualization:** Yin Mo, Ben S. Cooper.

**Writing – original draft:** Yin Mo.

**Writing – review & editing:** Yin Mo, Mathupanee Oonsivilai, Cherry Lim, Rene Niehus, Ben S. Cooper.

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
