## [Editor Report · Decision Letter 0]

4 May 2022

Dear Dr Mo, 

Thank you for submitting your manuscript entitled "The effect of reducing antibiotic treatment duration on antimicrobial resistance in the healthcare setting: a modelling study and meta-analysis" for consideration by PLOS Medicine.

Your manuscript has now been evaluated by the PLOS Medicine editorial staff and I am writing to let you know that we would like to send your submission out for external peer review.

Please re-submit your manuscript within two working days, i.e. by May 06 2022 11:59PM.

Kind regards,

Louise Gaynor-Brook, MBBS PhD

Senior Editor

PLOS Medicine

---

## [Decision Letter · Decision Letter 1]

11 Jul 2022

Dear Dr. Mo,

Thank you very much for submitting your manuscript "The effect of reducing antibiotic treatment duration on antimicrobial resistance in the healthcare setting: a modelling study and meta-analysis" (PMEDICINE-D-22-01395R1) for consideration at PLOS Medicine.

Your paper was discussed among the editorial team and we invite you to consider publication of your paper in PLOS Medicine's special issue entitled, " Bacterial antimicrobial resistance - surveillance and prevention". Please see the following link for further details: plos.io/AMR. 

If you are interested in your paper being published in the special issue please contact me on the email listed at the foot of this letter.

Your paper was sent to independent reviewers, including a statistical reviewer. The reviews are appended at the bottom of this email and any accompanying reviewer attachments can be seen via the link below:

[LINK]

In light of these reviews, we will not be able to accept the manuscript for publication in the journal in its current form, but we would like to invite you to submit a revised version that addresses the reviewers' and editors' comments fully. You will appreciate we cannot make a decision about publication until we have seen the revised manuscript and your response, and we expect to seek re-review by one or more of the reviewers. 

We hope to receive your revised manuscript by Aug 01 2022 11:59PM. Please email us (plosmedicine@plos.org) if you have any questions or concerns.

Please let me know if you have any questions, and we look forward to receiving your revised manuscript. 

Sincerely,

Philippa

Dr Philippa Dodd, MBBS MRCP PhD

Associate editor, PLOS Medicine

pdodd@plos.org

GENERAL

Please report your SR/MA according to the PRISMA 2020 guidelines provided at the EQUATOR site:

http://www.equator-network.org/reporting-guidelines/prisma/

Thank you for including the PRISMA diagram. Please also provide the completed PRISMA checklist. When completing the checklist, please use section and paragraph numbers, rather than page numbers.

We ask you to adapt the title to better match journal style, and suggest: "Implications of reducing antibiotic treatment duration for antimicrobial resistance in hospital settings: A modelling study and meta-analysis".

ABSTRACT

Please structure your abstract using the PLOS Medicine headings (Background, Methods and Findings, Conclusions).

Abstract Background: The final sentence should clearly state the study question.

Please combine the Methods and Findings sections into one section, “Methods and findings”.

Abstract Methods and Findings:

Please ensure that all numbers presented in the abstract are present and identical to numbers presented in the main manuscript text (i.e 187 studies Vs 7 included in analyses).

Please include the study design and main outcome measures.

Please include the important dependent variables that are adjusted for in the analyses.

In the last sentence of the Abstract Methods and Findings section, please describe the main limitation(s) of the study's methodology.

Please report your abstract according to PRISMA for abstracts, following the PLOS Medicine abstract structure (Background, Methods and Findings, Conclusions) http://www.plosmedicine.org/article/info:doi/10.1371/journal.pmed.1001419

Please provide the dates of search, data sources, number of studies included, types of study designs included, eligibility criteria, and synthesis/appraisal methods

Abstract Conclusions:

Please address the study implications without overreaching what can be concluded from the data; the phrase "In this study, we observed ..." may be useful.

Please interpret the study based on the results presented in the abstract, emphasizing what is new without overstating your conclusions.

After the abstract, please add a new and accessible 'Author summary' section in non-identical prose. You may find it helpful to consult one or two recent research papers in PLOS Medicine to get a sense of the preferred style. 

MAIN MANUSCRIPT

Please evaluate study quality and risk of bias.

Please evaluate evidence of publication bias.

Please add the following statement, or similar, to the Methods: "This study is reported as per the Preferred Reporting Items for Systematic Reviews and Meta-Analyses (PRISMA) guideline (S1 Checklist)."

Please clarify the discrepancy between the in the number of studies reported to be included in the analysis (7) and the number of studies referenced in the supp data (9) and provide an updated list as necessary

Please remove COI/funding source/data sharing info from the end of the manuscript and instead include in the manuscript submission form. 

Please also remove the 'Role of the funding source' statement from the end of the Methods section. 

At line 523, "... systematic review"?

Throughout the text, please style reference call-outs as follows: "... of resistance [5,6]." (noting the absence of spaces within the square brackets). 

In the reference list, please convert all italics to plain text. 

Where appropriate, please list 6 author names rather than 3, followed by "et al.".

FIGURES

Please provide titles and legends for all figures (including those in Supporting Information files). 

Please consider avoiding the use of red and green in order to make your figure more accessible to those with colour blindness

Comments from the reviewers:

*** Reviewer #1: 

This interesting manuscript describes a modelling study that examines the impact of duration of antibiotic treatment on antibiotic resistance in Gram negative bacteria using three different agent-based models. 

Strengths:

- The manuscript is well written (which is not an easy task given the complexity of the topic) with very thorough methodology

- Understanding the impact of duration of antibiotic therapy on antibiotic resistance is key to better inform antibiotic stewardship decisions

- The manuscript nicely highlights how complex the relationship between treatment duration and antibiotic resistance is and where the knowledge gaps are

Weaknesses:

- As with any model of this complexity a lot depends on the underlying assumptions which given the gaps in the literature remain speculative in many instances. The sensitivity analyses performed and the use of different model which give similar results are reassuring.

- At times this manuscript is difficult to flow which is probably mainly a consequence of the complexity of the subject. At times simple figures may help understanding. 

TITLE:

Minor comment: "The effect of reducing antibiotic treatment duration on antimicrobial resistance in the

healthcare setting" Hospital setting rather than healthcare setting is probably preferable?

ABSTRACT:

Major comment: While I understand that space is limited, I feel that the abstract could give more detail regarding the models and underlying assumptions. The way it is currently written it is difficult to eben get a vague idea about what was done.

Minor comment: The hypothetical setting should be described in the abstract.

Minor comment: The term "resistance carriage" should be better defined since it is the main outcome of interest

Minor comment: "7% increase in risk of resistance carriage" Absolute or relative?

INTRODUCTION:

Major comment: I think it would be good to add a paragraph in the introduction about the potential impact on transmission. The association of antibiotic treatment duration and length of stay (and thus probability of transmission / acquisition) would also be worth mentioning.

Major comment: It would also be important to mention that many of the resistant pathogens we care most about (e.g. carbapenemase and extended spectrum beta-lactamase producing Enterobacterales; carbapenem-resistant Acinetobacter) are not selected de novo, but rather acquired through transmission (Unlike fluoroquinolone resistance in Enterobacterales, one will not "create" a carbapenemase producing E. coli even when giving carbapenems for months). I understand that this intended by the "bystander" terminology but I think it would be good to give concrete examples. 

Minor comment: "Examples of pathogens which may become resistant with inappropriate treatment include Mycobacterium tuberculosis and human immunodeficiency virus." Rather than using HIV as an example I would mention fluoroquinolone resistance in Enterobacterales.

METHODS AND MATERIALS:

Major comment: It would be good to describe the setup of the modelled hospital in more detail (I assume transfer between units was not simulated?)

Minor comment: "Within the same ward, antibiotics were prescribed with the same mean duration on or during admission." This is obviously quite an oversimplification and should be mentioned as a limitation. Why were the values not sampled from a distribution?

Minor comment "Real-world parameter values were obtained from the literature and used in these explorations (Supplementary material 1 Table 1.2)." These parameters are so essential for the interpretation of the model that they should be part of the main text. Also, some parameters have no references and it should be explained how they were derived.

Minor comment: Was the systematic review registered?

RESULTS:

Minor comment: "Key parameters in the relationship between antibiotic duration and resistance carriage" This section is quite difficult to understand. I acknowledge that is difficult to describe these complex findings simply, maybe some illustrations could help?

DISCUSSION:

Major comment: It should be mentioned as a limitation that both antibiotic use and resistance is much more complex than modelled here (many different antibiotics often in combination for different durations and routes of administrations and dosages over multiple courses with multiple different resistance pathogens with different resistance mechanisms)

Minor comment: "The study by D'Agata et al" This study seems to be missing from the references, idem Blanquart et al. It seems to me that there are two modelling papers by D'Agata looking at treatment duration and resistance (interestingly with somewhat opposing conclusions)

https://journals.plos.org/plosone/article?id=10.1371/journal.pone.0004036

https://www.ncbi.nlm.nih.gov/pmc/articles/PMC2432019/

FIGURES AND TABLES:

Minor comment: I found figure 1 not simple to understand. Care should be taken to consider readability. The legends of some of the other figures were also difficult to follow (again I acknowledge that this is partly a consequence of the complexity of the topic and models).

REFERENCES:

Minor comments: Some references are not formatted. 

Minor comment: There are now also some studies using metagenomics, that show that the impact of treatment duration on intestinal resistance genes for examples is far from obvious https://pubmed.ncbi.nlm.nih.gov/34492446/

*** Reviewer #2 (methodological reviewer): 

"The effect of reducing antibiotic treatment duration on antimicrobial resistance in the healthcare setting: a modelling study and meta-analysis" examines said effect in two largely-separate ways: firstly, through mathematical modelling, and secondly through a meta-analysis, which yielded five suitable prior randomised trials. Both methods suggest that reduced antibiotic treatment duration can result in some reduction in resistance carriage. These findings appear plausible, although there are some reservations relating to the representativeness of the mathematical models, and the relatively small sample size of the meta-analysis. Detailed comments follow:

1. It might be clarified as to whether the three models presented (Figure 1) are standard/known from prior work, since there appears extant literature on mathematical/computational modelling for colonisation (resistance), e.g. Boureau, L. Hartmann, T. Karjalainen, I. Rowland, MHF Wilkinson, H. (2000). Models to study colonisation and colonisation resistance. Microbial Ecology in Health and Disease, 12(2), 247-258.

2. In Supplementary S1.2, it is observed that many of the parameters do not appear to have relevant references cited in support of their chosen value ranges. For these parameters, the chosen ranges might be briefly justified.

3. In Supplementary S1.3.1, Figure S2 appears to be labeled as S1. This might be addressed, together with its corresponding reference(s) in the text.

4. In S1.3.2, it is stated that "Non-monotonicities in the correlations between the parameters and the model output were examined visually through scatter plots and calculating the Hoeffding's D measure and Spearman's rank correlation measure"; were any such non-monotonicities found, addressed further?

5. Figure S3 is stated to show scatter plots of the parameters as sampled from hypercubes formed within each parameter space. Then, it is assumed that for each model output (resistance/non-resistance carrier value), each point within the scatterplot for some input variable (e.g. n) represents the estimated relationship between that input variable and the output, with the remaining input variables values being randomly sampled. It is further assumed that each point derives from the "averaged model outputs from [100 iterations] for the same set of parameter combination inputs", as described in S.1.3.1.

If the above understanding is correct, it might be justified as to whether the 500 points (i.e. independent sets of [different] parameter combination inputs, from Line 190) that were used in the scatterplots and associated regression estimates is sufficient. This is because the number of (input) parameter combinations increases exponentially with the number of parameter (i.e. the "curse of dimensionality"). For example, if we consider that a parameter is merely binary (i.e. has a high or low value only), and there are 16 such parameters (as in the smallest exclusive colonisation model), then simply representing all binary input combinations would take 2^16=65,536 points.

An empirical method of testing this might be to re-run the experiments with a different seed/set of 500 unique sets of parameters), and to observe whether similar conclusions are obtained (as presented in Figure 4). This might be warranted as the effects appear minimal for most variables, although there does seem a consistent pattern for p_R on resistant carriers in overall ward population across all three models for example (which however seems somewhat tautological).

6. The caption of Figure S3 further states that the lines in the scatterplots are regression lines obtained from locally estimated scatterplot smoothing. The algorithm/procedure for plotting these lines from the points might be briefly described.

7. Tradeoffs associated with shortening antibiotic treatment duration might also be acknowledged and discussed, as otherwise a reduction to zero duration would appear optimal.

Minor issues:

(Line 78) "This selective pressure for resistance most obviously applies..." might be considered to be rephrased (perhaps as "most directly applies")

(Line 270) "then acted as a reservoirs" might be "reservior"

*** Reviewer #3: 

Many thanks for the opportunity to review this interesting and well written manuscript by Mo and colleagues. The manuscript aimed to explore the relationship between antimicrobial duration and colonisation with resistant Gram-negative bacteria in a hospital setting. They aimed to compare the findings from reported models to observations reported in the literature by performing a systematic review and meta-analysis. 

This study highlights the challenges of trying to determine the impact of any single intervention (in this case shortening treatment duration) on carriage / propagation of AMR. It approaches this in a logical and open way, acknowledging the limitations of the models used and attempting to triangulate observations where possible. It also reinforces many of the challenges we have in design and assessment of impact of stewardship programmes on AMR. A critical point is who, how often, and where we should be following up patients in trials to try and determine the true impact of reducing antimicrobial consumption on AMR (at the patient, ward, even regional level). 

In terms of the models used and moving forwards, it would be interesting to understand whether these could now be applied to a population level, where we often consider antimicrobial consumption and the impact on AMR. For example, modelling a hospital using individual ward data predictions to highlight wards that are safe for stewardship interventions / focus on trying to reduce antimicrobial prescribing. Could it also better inform optimal times for sampling in these studies?

For the systematic review, the methodology is a little light. 

Was the manuscript registered on prospero or similar prospective registry prior to starting?

Was any assessment for risk of bias performed? Were the PRISMA reporting guidelines followed for systematic review and meta-analysis?

***

[LINK]

---

## [Decision Letter · Decision Letter 2]

2 Sep 2022

Dear Dr. Mo,

Thank you very much for re-submitting your manuscript "Implications of reducing antibiotic treatment duration for antimicrobial resistance in hospital settings: A modelling study and meta-analysis" (PMEDICINE-D-22-01395R2) for review by PLOS Medicine.

I have discussed the paper with my colleagues and the academic editor and it was also seen again by 4 reviewers. I am pleased to say that provided the remaining editorial and production issues are dealt with we are planning to accept the paper for publication in the journal.

[LINK]

We look forward to receiving the revised manuscript by Sep 09 2022 11:59PM.   

Sincerely,

Philippa Dodd, MBBS MRCP PhD

Senior Editor 

PLOS Medicine

pdodd@plos.org

plosmedicine.org

Requests from Editors:

Please address all additional reviewer comments below

Throughout, please ensure consistent use of terminology (e.g. antibiotic Vs agent)

Throughout, please ensure consistent use of words (five) or numbers (5) when presenting/discussing numerical values

Abstract

Thank you for adding the following sentence to the abstract “The meta-analysis was performed with a logistic regression ….” Perhaps reword for improved clarity, suggest “The meta-analysis was performed using logistic regression. Duration of antibiotic treatment and time from administration of antibiotics to surveillance culture were included as independent variables” or something similar, as we understand it.

Thank you for adding the following sentence, “Interpretation of these estimates is limited by the low number of antibiotic duration trials which monitored resistance carriage as an outcome, contributing to a large credible interval.” Suggest modifying to “…which monitored carriage of resistant gram negative bacteria, as an outcome…” for clarity and consistency.

Author Summary

Thank you for including an author summary. Please ensure consistent terminology is used throughout – suggest either antibiotic treatment or antibiotic duration or antibiotic treatment duration but not different terms, similarly for antibiotic Vs agent. Some further suggestions are below:

Why was this study done?

* Suggest revising the following sentence to “…this depends on the agent’s spectrum of coverage, pharmacokinetic and pharmacodynamic properties…” – as bioavailability forms part of the aforementioned.

* Consider revising the sentence beginning “Empirical evidence…” to “Empirical evidence and an understanding of the mechanisms by which antibiotic treatment duration effects the emergence and spread of antimicrobial resistance are lacking.” or something similar

* “…factors driving the effect [of] antibiotic…”? consider also adding the words “…[help to] inform…”

What did the researchers do and find?

Some of these statements appear repetitive. Please refine this section to include the studies primary findings.

* Please add [antibiotic treatment] or something similar to the following sentence “The models showed that shortening [ ] duration is most effective at reducing resistance carriage in high transmission settings”

What do these findings mean?

Please revise this section and include two or three main implications of the study findings that you list above. 

Some of the points currently in this section might better placed above - some more specific comments are below:

* “Shortening antibiotic duration…” consider revising to “…Shortening the duration of antibiotic treatment may increase or decrease colonization by resistant bacteria depending upon the effectiveness of antibiotics” or something similar.

* Please clarify the following statement “For treated individuals, shortening duration is most effective when resistant bacteria grow rapidly under antibiotic selection pressure and decline rapidly when stopping treatment.” Most effective at reducing resistance? 

* The following statement - “The modest effect of shortening treatment duration on resistant bacteria colonisation can be partially explained by typically short hospitalisation stays and a slow rate of decolonisation of resistant bacteria in patients previously exposed to antibiotics” is long and difficult to read/understand. Please either revise to improve accessibility to the reader or remove 

* Consider the following sentence, or something similar in place of the current “We found that an important interplay exists between shortening antibiotic treatment and reducing the transmission of resistant bacteria through infection prevention and control” as we understand it

* Please remove the statement on study limitations

Methods & Findings

Thank you for including assessment of the individual risk of bias for each study included in the meta-analysis. 

The assessment of publication bias doesn’t appear to be included. While it is good practice to include a minimum of 10 studies when assessing risk of bias using tests for funnel plot asymmetry (if this underpins the omission?), alternative strategies, such as Tang’s regression test (PMID: 29663281) could be an alternative approach in respect of your meta-analysis which includes 5 studies. You may find further guidance here PMID: 10812319 helpful also. An additional comment on the limitations of assessment of publication bias in your study (perhaps in the discussion section of the main manuscript) as appropriate, would also be beneficial.

Figures

Thank you for your attention to our requests regarding the figures. 

It appears that throughout, the figures are now rather out of sink with the figure captions. For example line 162, Figure 1…the figure is below the figure caption. The same is seen with all subsequent figures in the main manuscript. Throughout, please place the figures above the figure caption and ensure that captions are clearly distinct from the main text.

Thank you for altering the colour schemes in figure 1. It might be advisable to opt for colours that are very distant from red and green - the pink shading used in figure 1 appears rather red, perhaps varying shades of light and dark blue, for example, or something similar. In addition it is difficult to read the white letters overlying the yellow background so suggest amending for improved readability.

Figures 2 and 3 (heatmaps) are unchanged in colour – perhaps because re-analysis may be required? If they can be changed then please do so otherwise please indicate in your response if and where colour schemes cannot be altered for any reason so that I cease to pester you about it! 

Throughout the manuscript, if/when figure colors are altered, please also remember to amend the descriptive text e.g. line 387 in reference to figure 4 “Red, blue and grey coloured rectangles…”

References

Ref #9 please spell out the names of the first 6 authors names

Please ensure all web references have an access date e.g. ref #2

*** If you have any specific questions or require clarification or further assistance please do not hesitate to contact me on the personal email address detailed in this letter ***

Comments from Reviewers:

Reviewer #1: The authors have addressed all my major comments. I congratulate them for this important work.

Reviewer #2 (methodological reviewer): We thank the authors for largely addressing our previous comments. A few minor suggestions might be considered:

1. While 10 repeats for various parameter sets were attempted, the concern about the curse of dimensionality appears to still possibly apply to unexplored combinations of parameters. This might be briefly mentioned as a potential limitation.

2. For the loess, the relevant hyperparameters used for fitting might be included, if relevant.

3. While tradeoffs for optimizing antibiotic treatment duration are now briefly discussed, the condition of "...optimising definitive antibiotic courses for established bacterial infections while ensuring clinical cure, and rapid discontinuation of empiric prescriptions after bacterial infections" appears fairly vague. Have any quantitative guidelines for the tradeoffs been considered?

Reviewer #3: Authors have addressed my previous comments. 

No additional comments to add. 

Many thanks and congratulations. 

Reviewer #4: The authors have addressed the comments I have provided earlier and improved the overall quality of the manuscript significantly. The methodology section has been clarified, especially around the assumptions made to conceptualise the agent based model. The steps taken to simulate the model have now been clearly described to allow audience who were unfamiliar with the method to understand the approach. I believe the revised manuscript fits for publication and have no further comments.

[LINK]

---

## [Decision Letter · Decision Letter 3]

3 Oct 2022

Dear Dr. Mo,

Thank you very much for re-submitting your manuscript "Implications of reducing antibiotic treatment duration for antimicrobial resistance in hospital settings: A modelling study and meta-analysis" (PMEDICINE-D-22-01395R3) for review by PLOS Medicine.

I have discussed the paper with my colleagues and the academic editor and it was also seen again by xxx reviewers. I am pleased to say that provided the remaining editorial and production issues are dealt with we are planning to accept the paper for publication in the journal.

[LINK]

We look forward to receiving the revised manuscript by Oct 10 2022 11:59PM.   

Sincerely,

Philippa Dodd, MBBS MRCP PhD

PLOS Medicine

pdodd@plos.org

plosmedicine.org

Requests from Editors:

Please accept my apologies for the delay in responding to you, we are unexpectedly very short staffed currently. Thank you for addressing the previous editorial and reviewer comments. There are some further minor revisions as detailed below that are remain outstanding. If you have any questions related to these please contact me directly via email (pdodd@plos.org) I am happy to discuss via email to expedite the process for you as much as I can.

ABSTRACT

Line 48: “Importantly, shortening antibiotic treatment may increase resistance carriage when antibiotics administered can effectively suppress the colonising bacteria with a particular resistance phenotype….”, is rather confusing to me. I have suggested a revision, below, as per my understanding – please correct this if I am mistaken! “Importantly, under circumstances whereby administered antibiotics can suppress colonizing bacteria, shortening antibiotic treatment may increase the carriage of a particular resistance phenotype.”

Please also amend the abstract in the manuscript submission form when you resubmit 

Line 50-53: beginning we “…We identified…” suggest revise to the following as per my understanding:

“We identified 187 randomised trials which investigated antibiotic duration. Of these, 5 reported resistant Gram-negative bacteria carriage as an outcome and were included in the meta-analysis. The meta-analysis determined that a single additional antibiotic treatment day is associated with a 7% absolute increase in risk of resistance carriage (80% credible interval 3 to 11%).”

Notably, the information in the abstract and main manuscript differ:

Line 492: the section titled “evidence from randomized controlled trials” will need revising. 

Line 493: “Initial search of the MEDLINE and EMBASE databases returned 2407 unique publications. Out of these 187 were randomised trials which compared antibiotic treatment durations. 7 these trials collected surveillance cultures for colonising bacteria during follow-up visits and were included in the qualitative synthesis.” However in the abstract report the inclusion of 5 studies…. 

Only later do we find out that a further 2 of these studies were excluded which is a bit confusing. For clarity and transparency the information from line 512 should be reported earlier, at least before we are directed to the table of study characteristics which reports 5 not 7 studies. 

Table 1 also only details 5 studies rather than 7…please also update the PRISMA flow charts for study exclusion as necessary 

In the abstract, you report the search was performed up to May 2021, can you please update it to within the last 6 months. I suspect there will not be any additional trials to include but please update the search dates in any case and the relevant PRISMA flowcharts etc. 

Please include the updated search dates in the main manuscript text where you describe the metanalysis and systematic review (line 277). Please ensure any updates are also made to supplementary files where relevant

AUTHOR SUMMARY

This should be concise, distinct from the abstract and accessible to the lay reader. I have revised, as per my understanding, as below:

Why was this study done?

• Shortening antibiotic treatment duration is a commonly adopted antibiotic stewardship strategy, with the expectation that it will reduce antimicrobial resistance in treated individuals and in the overall population. 

• Antibiotic selective pressure acts predominantly on ‘bystander’ colonising bacteria for resistance, and this depends on the spectrum of coverage, pharmacokinetic and pharmacodynamic properties of individual antibiotics.

• Empirical evidence and an understanding of the mechanisms by which antibiotic treatment duration effects the emergence and spread of antimicrobial resistance are lacking. Understanding the key factors driving the effect of antibiotic treatment duration on resistance carriage will help to inform future research study designs, antimicrobial stewardship interventions and resource allocation in the multimodal control strategies. 

What did the researchers do and find? 

• We modelled within- and between-host dynamics of colonising ‘bystander’ susceptible and resistant bacteria in response to systemic antibiotic treatment and compared the model findings with a systematic review and meta-analysis. 

• The meta-analysis found one additional antibiotic treatment day is associated with a 7% absolute increase in risk of resistance carriage when antibiotics administered were not effective against the resistance phenotype in the colonising bacteria. 

• For treated individuals, the models showed that shortening antibiotic treatment duration is most effective at reducing resistance carriage when resistant bacteria grow rapidly under antibiotic selection pressure and decline rapidly when stopping treatment.

 • At a population level, shortening antibiotic treatment duration is most effective at reducing resistance carriage in high transmission settings. 

• Shortening antibiotic treatment duration may increase resistance carriage when the antibiotics administered are effective at eliminating colonising bacteria with a particular resistance phenotype. 

What do these findings mean? 

• Shortening antibiotic treatment duration may increase or decrease colonisation by resistant bacteria, dependent upon individual and combined bacterial and antibiotic characteristics, and may help to inform antimicrobial stewardship policy making. 

Please amend if I am mistaken in my interpretation at any point in editing this summary

GENERIC INFORMATION FOR MODELLING STUDIES

Please check the list below and ensure details are adequately detailed in the manuscript 

Please provide a diagram that shows the model structure, including how the disease natural history is represented, the process and determinants of disease acquisition, and how the putative intervention could affect the system.

Please provide a complete list of model parameters, including clear and precise descriptions of [the meaning of each parameter, together with the values or ranges for each, with justification or the primary source cited, and important caveats about the use of these values noted].

Please provide a clear statement about how the model was fitted to the data [including goodness-of-fit measure, the numerical algorithm used, which parameter varied, constraints imposed on parameter values, and starting conditions].

For uncertainty analyses, please state the sources of uncertainties quantified and not quantified [can include parameter, data, and model structure].

Please provide sensitivity analyses to identify which parameter values are most important in the model. Uncertainty estimates seek to derive a range of credible results on the basis of an exploration of the range of reasonable parameter values. The choice of method should be presented and justified.

Please discuss the scientific rationale for this choice of model structure and identify points where this choice could influence conclusions drawn. Please also describe the strength of the scientific basis underlying the key model assumptions.

REFERENCES

Line 569: “…community were used.[39]…” this error is repeated throughout the manuscript, please check and amend throughout as per below:

Please select the PLOS Medicine reference style in your citation manager. Please use square brackets for in-text reference call outs noting the absence of spaces within the square brackets, the space before the first bracket and the punctuation following the second bracket, “…symptomatic [2,8].”

PLOS uses the reference style outlined by the International Committee of Medical Journal Editors (ICMJE), also referred to as the “Vancouver” style. Example formats are listed below. Additional examples are in the ICMJE sample references. Please list up to 6 author names only before et al where more than 8 authors have contributed

Example: Hou WR, Hou YL, Wu GF, Song Y, Su XL, Sun B, et al. cDNA, genomic sequence cloning and overexpression of ribosomal protein gene L9 (rpL9) of the giant panda (Ailuropoda melanoleuca). Genet Mol Res. 2011;10: 1576-1588.

Comments from the Guest Editors:

it's a nice piece of work and I hope you will go ahead and accept it if not already.

Comments from Reviewers:

Reviewer #2: We thank the authors for addressing the previous comments as best as possible, and have no further issues to raise.

[LINK]

---

## [Editor Report · Decision Letter 4]

17 Oct 2022

Dear Dr Mo, 

On behalf of my colleagues and the Academic Editor, Professor Ramanan Laxminarayan, I am pleased to inform you that we have agreed to publish your manuscript "Implications of reducing antibiotic treatment duration for antimicrobial resistance in hospital settings: A modelling study and meta-analysis" (PMEDICINE-D-22-01395R4) in PLOS Medicine's Antimicrobial Resistance and Surveillance Special Issue.

There is a single, final revision detailed below for you to make prior to publication. It has been a pleasure handling your manuscript throughout the peer review and editorial process, many congratulations on its publication.

Comments from the Editor:

Line 279: "2000 up to 20 May 2021..." you updated your search and amended the search dates in the abstract and supplementary material but not in the main manuscript, please amend accordingly.

PRESS

Best wishes,

Pippa 

Philippa Dodd, MBBS MRCP PhD 

Editor 

PLOS Medicine

pdodd@plos.org